# Development of an Energy Efficient and Cost Effective Autonomous Vehicle Research Platform

**DOI:** 10.3390/s22165999

**Published:** 2022-08-11

**Authors:** Nicholas E. Brown, Johan F. Rojas, Nicholas A. Goberville, Hamzeh Alzubi, Qusay AlRousan, Chieh (Ross) Wang, Shean Huff, Jackeline Rios-Torres, Ali Riza Ekti, Tim J. LaClair, Richard Meyer, Zachary D. Asher

**Affiliations:** 1Western Michigan University, 1903 W Michigan Ave., Kalamazoo, MI 49008, USA or; 2FEV North America Inc., 4554 Glenmeade Ln, Auburn Hills, MI 48326, USA; 3Oak Ridge National Laboratory, 1 Bethel Valley Rd., Oak Ridge, TN 37831, USA

**Keywords:** autonomous vehicle system, connected and automated vehicle, self-driving cars, sensors, perception, radar, camera, LiDAR, obstacle detection, intelligent transportation system

## Abstract

Commercialization of autonomous vehicle technology is a major goal of the automotive industry, thus research in this space is rapidly expanding across the world. However, despite this high level of research activity, literature detailing a straightforward and cost-effective approach to the development of an AV research platform is sparse. To address this need, we present the methodology and results regarding the AV instrumentation and controls of a 2019 Kia Niro which was developed for a local AV pilot program. This platform includes a drive-by-wire actuation kit, Aptiv electronically scanning radar, stereo camera, MobilEye computer vision system, LiDAR, inertial measurement unit, two global positioning system receivers to provide heading information, and an in-vehicle computer for driving environment perception and path planning. Robotic Operating System software is used as the system middleware between the instruments and the autonomous application algorithms. After selection, installation, and integration of these components, our results show successful utilization of all sensors, drive-by-wire functionality, a total additional power* consumption of 242.8 Watts (*Typical), and an overall cost of $118,189 USD, which is a significant saving compared to other commercially available systems with similar functionality. This vehicle continues to serve as our primary AV research and development platform.

## 1. Introduction

Academia, private, and government entities have been developing autonomous navigation systems for sea [1], land [2], air [3], and space [4] to improve safety and efficiency for decades. To date, robust systems have been achieved in each of these domains with the glaring exception of on-road autonomous vehicles (AVs) due to the greater complexity of the driving environment [5]. Although no robust AV system is currently produced with fully autonomous capabilities, there is still a substantial amount of ongoing AV research [6,7], and in order to successfully transition these research efforts into commercially available real-world products, an AV research platform is needed to further acceleration AV applied research. Robustness can be said to be in accordance with the ADAS standards and ADAS safety protocols which are set by SAE and ISO [8], yet ADAS does not cover full AV vehicles since they are technically not at that point, yet are working closely to developing a safe compliant AV. Successful operations of any AVs [9,10] for the aforementioned research platform strictly depend on the orchestration of computer vision systems and multiple sensors such as MobilEye, radar, LiDAR [11], Global Navigation Satellite System (GNSS), Inertial Measurement Unit (IMU) [12], and a high-powered computer capable of running complex sensor processing algorithms. Data is often collected from these sensors via the Controller Area Network (CAN) bus connection (i.e., serial cable or On-Board Diagnostics (OBD) port) from the Ego vehicle embedded system, Drive-Kit, radar, and MobilEye. The LiDAR and GNSS/IMU sensors normally communicate data over ethernet connections, whereas the camera uses USB 3.0. Automotive Original Equipment Manufacturers (OEMs) often utilize their own internally designed and proprietary systems to connect with all of these sensors, whereas Tier 1 suppliers, academic groups, and others typically employ free and open-source software such as the Robotic Operating System (ROS) [13,14,15]. Initial literature on this topic shows significant success (i.e., rapid prototyping, global and local path planning, and localization) with AV system integration when using ROS [16,17,18]. By having a universal timestamp for when all of the sensors are functioning in real-time, ROS will be used to maintain track of sensor data. This will aid with data analysis and timesync tracking throughout the Ego vehicle environment.

There are a few research papers that detail development of custom AV systems such as [19,20,21]. Baxter’s study focuses on the sensor electrical power requirement of an AV [19]. Azam’s study shows how to operate a fully functional autonomous taxi, while using a minimum amount of sensor instrumented on the vehicle [20]. Although their system works, the camera selection causes a problem during inclement weather, and they do not mention the cost of instrumenting their car. Walling’s thesis discusses an overview of the design of AV showing different platforms, yet does not discuss the power consumption of any platform or the cost of the sensor suite cost [21]. We used these studies as a reference on how (1) the sensors should be placed for effective perception of the Ego-vehicles environment [21,22], (2) how much power consumption should a typical sensor suite have [19,23], which ranges between 200 to 300 Watts (note: high performance GPU can increase these number drastically), and (3) the minimum sensors needed to have a fully functional level 4 AV platform [24]. There are vendors willing to supply equipment and prices, but due to fluctuating prices and a competitive market, these prices remain confidential. We hope to benefit the community by sharing the results of our AV development.

To create a cost-effective AV development platform, strategic decisions must be made in vehicle selection, drive-by-wire (DBW) functionality, an effective sensor package [9], and system architecture [16,25]. This can be a confusing and time-consuming task for which one might consult the literature as a guide. For example, discussed hardware for computer vision object detection using a ROS architecture to achieve path planning [16,17]. Others review perception system design and AV perception limitations [26] where it is discussed that the capabilities and limitations of using Advanced Driver Assistance Systems (ADAS) are still a safety concern [27]. Another study provides an informative overview of sensors and vehicle systems, describing how they can be used for various functionalities and the advantages and disadvantages of specific devices for enabling AV operation which was taken into account [28]. Despite these references, there is a considerable research gap in terms of specific studies that address sensor selection, performance, and, more crucially, power consumption, as well as cost considerations.

We propose a comprehensive and cost-effective approach created by a joint academic and industrial team while instrumenting an AV for pilot operation testing in Detroit, MI, to fill this research gap. We also document the power usage of each new component to facilitate AV energy efficiency assessments. This is essential, since new issues about AV energy efficiency are emerging, and researchers must now address the long-term ramifications of created solutions. This method is meant to help a wide spectrum of researchers improve the transition from laboratory and simulation work to real-world demonstrations and deployments. We demonstrate how to convert a 2019 Kia Niro Hybrid electric vehicle (HEV) into a cost-effective and energy-efficient physical testing platform for AV development, which is necessary since the sensor suite increases the load on the battery, which increases the rate at which the battery discharges. This shows how there is a need for energy efficiency. This procedure may be standardized, discussed with the research community, and completed at a cheap cost by utilizing free and open-source software such as ROS and Linux [14], which ultimately accelerates AV realization and benefits for society [25].

### Organization of the Remainder of the Paper

Section 2 details the methodology for system architecture, equipment selection, and integration. Results and relevant discussions regarding sensor operation, total system cost, and power consumption are presented in Section 3. Finally, discussions and concluding remarks are drawn in Section 3.5 and Section 4, respectively.

## 2. Methodology

To match the traditional AV platform [25], sensor placement [22,28], and computer architecture in the literature [16,29], we have elected to use a commonly available vehicle, commercially available DBW kit, a stereo camera, the MobilEye computer vision product, radar, LiDAR, a combination GPS/IMU, and a high-performance computer. Specifically, we chose the PolySync Drive-Kit [30,31], a ZED stereo camera, MobilEye 630, Aptiv ESR, Ouster OS1 LiDAR, two Swift Navigation Duro GNSS/IMUs, and a Nuvo-7160GC ruggedized PC that runs Linux Operating System (OS). Details of the selection, instrumentation, and outputs of each of these sensors and system components are presented in subsequent subsections.

The sensors and hardware that were considered as potential sensors for the vehicle are shown in Table 1, along with a comparison of those sensors to other sensors based on the average of the data collected. The open source data indicates that ROS drivers, data sheet specifications, and articles are generally available. When determining compatibility, the system’s overall connectivity to the vehicle, sensors, and AV computing systems is taken into account. The average of each individual sensor group was established based on the type, capabilities, and specifications of the sensor as well as other similar sensors. The average of each sensor group is used to calculate that group’s ranking for open source, capability, cost, and power consumption.

### 2.1. Vehicle Selection

Autonomous operation of an automotive vehicle first requires DBW capability. DBW is the road vehicle equivalent of the fly-by-wire systems originally implemented in airplanes [32,33], and it implies that vehicle control inputs (e.g., steering wheel, throttle, brake pedal, etc.) can be wholly operated by electronic inputs such as a digital signal, rather than only mechanical inputs such as rotating a steering wheel. The first step in the development of an AV research platform is to choose a vehicle that is DBW capable. We chose a 2019 Kia Niro Hybrid Electric Vehicle (HEV), shown in Figure 1. This vehicle is DBW compatible since it is designed to have electric actuation of acceleration, braking, and steering. For autonomous integration, the vehicle’s CAN bus is accessible and is used to send and receive individual sensor signals [25,34,35].

Planning of sensor placement also influences vehicle selection. AV instrumentation engineering requirements for sensor placement include front-facing cameras, a front-facing radar, a 360-degree field of view for the LiDAR, and externally mounted GNSS/IMU for unobstructed communication [36]. Figure 2 shows our chosen placement of sensors on the vehicle, which adheres to all engineering requirements and is also aligned with decisions made by other researchers [12,22,37]. The high-performance computer and its associated hardware (inverter, power distribution box, fuse box, and solenoid) are stored in the rear cargo area to be easily accessible. A diagram of these equipment locations is also shown in Figure 2 and all of these are achievable on the 2019 Kia Niro. Additionally seen in Figure 2, the radar is placed on the front bumper for oncoming detections such as pedestrians and traffic. The stereo camera was placed in the centerline on the windshield. The MobilEye was centered on the front windshield similar to the stereo camera for accurate lane and object detection [38]. The LiDAR and GNSS/IMU were placed on top with both sensors along the centerline of the vehicle for the heading of the Ego vehicle [13,39]. The LiDAR can be seen in Figure 2, mounted externally in the center of the vehicle to achieve a full 360-degree view. This will provide ground truth for objects detected around the Ego vehicle [40,41,42].

### 2.2. DBW Interface: Polysync Drive-Kit

A subsystem to convert driving commands from the in-vehicle computer to the vehicle actuators through the CAN bus is also needed. We chose Polysync’s Drive-Kit product since it was designed so that any driver commands inherently override the Drive-Kit commands for safety. All vehicle actuation commands are sent as native CAN signal IDs and do not actuate through, but in parallel with, ADAS products such as lane-keeping assistance on a separate network other than the OEM network. This approach is also safer in our opinion, since the vehicle CAN bus will not be flooded with large amounts of data to its network, which in turn causes congestion of information. In practice, we have found that installation of the DBW interface takes about 30 minutes, has a sound operation, and is highly intuitive for users. Flooding of data through the OEM CAN bus can cause lag in controls and could cause injury. Components of Polysync’s Drive-Kit can be seen in Figure 3.

The Drive-Kit includes a ROS driver from Github [44,45] to connect to the control module to send and receive vehicle messages. The harness plugs are in parallel with the vehicle harness system that matches the OEM connections. The computer for the Drive-Kit is under the front driver seat to maintain stock appearances while still allowing easy access, and has a measured minimum power consumption of 2.7 Watts. It also includes a Kvaser cable for interfacing with the in-vehicle computer. The Kvaser cable was routed to the rear along the driver side of the vehicle hidden under the trim.

### 2.3. Camera Sensor: ZED by StereoLabs

When choosing a camera sensor, we chose to prioritize high levels of functionality as well as ease of use to ensure utility in future projects. The ZED stereo camera developed by Stereo Labs provides high-resolution images as well as depth information. StereoLabs also provides a Software Development Kit (SDK) that allows the user to interface with the camera using their choice of Python or C++. This SDK is a major benefit for our AV research platform because it enables straightforward development of object detection, object tracking, Simultaneous Localization and Mapping (SLAM), lane line detection, and more [46].

Additionally, if ROS is being used as the middleware for the AV software stack, there is a ROS driver available [47] that facilitates straightforward system integration. Using ROS, the outputs from the sensors and custom algorithms can be accessed by subscribing to the topics that are published from the StereoLabs ROS driver. An output example for the ZED camera can be seen in Section 3.1. The camera can be mounted on the vehicle’s dashboard using commonly available mounting brackets as shown in Figure 4. After installation, we were able to quickly and easily record high-quality camera images. Calibration is needed to have more accurate data for localization and object detection; ZED has clear instructions on how to calibrate and use their software development kit (SDK). The specification of the stereo camera can be found on their website for further investigation [48].

### 2.4. Camera Computer Vision Product: MobilEye 630

For a commercial off-the-shelf (COTS) computer vision (CV) product, we selected a proven system that is currently used by many OEMs to accurately detect road features and object attributes [49]. The COTS computer vision product we chose is the MobilEye 630, which provides detections for lane lines, vehicles, pedestrians, motorcyclists, cyclists, signs, traffic lights, and more. This sensor also provides the speed of and distance to the vehicle in front of the AV. Additionally, it provides warning sounds that can be used in driver assistance applications. Because of these out-of-the-box capabilities, many automotive OEMs and Tier 1 suppliers include this product in their AV platforms [22,25,27]. MobilEye software drivers are available [29] that can be installed to work with ROS. These drivers enable a straightforward implementation of this sensor. An example output from the MobilEye can be seen in Section 3.1.

Note that this MobilEye sensor was installed and calibrated by a MobilEye certified/authorized technician, which is recommended for installation. If installed without a technician, the MobilEye is not guaranteed to be accurate. The sensor’s placement is on the center of the vehicle’s front windshield, as shown in Figure 5. The MobilEye power is controlled by the ignition switch while the data is transmitted using CAN signals. This connection is routed along the driver-side interior trim to the Kvaser leaf (CAN to USB) that is connected to the high-performance PC in the rear of the vehicle. This was one of two sensors purchased from the AutonomouStuff dealer. The specifications of the MobilEye 630 camera can be found on AutonomouStuff’s website for further investigation [50].

### 2.5. Radar: Aptiv ESR 24 V

When choosing a radar system, we chose to prioritize an electronically scanning radar (ESR) with high accuracy, durability, and consistency with other research [51,52]. The Aptiv ESR 2.5 24 Volt sensor provides a combination of long-range and mid-range radar sensing, which enables concurrent measurements for forward-facing vision, as shown in Figure 6. More detailed information can be found in the Aptiv ESR documentation provided publicly from AutonomouStuff [53]. The radar detects up to 64 objects’ positions, direction, and relative speeds for each scan, which is useful for congested city and highway driving applications. It will detect static objects such as guard rails and dynamic objects such as pedestrians and vehicles of any size. The radar memory is low enough to help conserve computational complexity as well as have enough memory to detect objects for ACC safely. These radars are currently being used for autonomous taxis in many cities [54]. Note that we initially selected a Bosch mid-range radar (MRR), but it was replaced with the Aptiv ESR due to a lack of sound ROS drivers and Database Container (DBC) file which helps decode CAN bus data using C++/Python as an alternative to ROS drivers. The Aptiv also comes with software that can be viewed on Windows OS to also retrieve and visualize the data. The Aptiv ESR has ROS drivers online that can be found on the AutonomouStuff ROS Wiki website for easy software installation [55]. An output example for the ESR can be seen in Section 3.1.

The Aptiv ESR includes a mounting bracket for calibrating the correct orientation angle. This bracket needs an additional fabricated bracket that attaches to the front structural bumper (not to be mistaken for the bumper cover of the vehicle). The sensor must be forward-facing and placed, so that the center of the radar is also horizontally aligned to the centerline of the car, shown in Figure 6 and Figure 7. The vertical distance of the radar from the driving surface should be between 350 mm and 600 mm. The adjustment bracket that came with the Aptiv radar can be mounted to the vehicle’s front bumper, as shown in Figure 7. This bracket can be used for other vehicles with similar bumper placement. The harness was routed through the air dam, shown in Figure 7, through the engine bay, and through an existing hole in the firewall where all the OEM harnesses enter the cabin of the vehicle. From there, the harness was routed along the driver-side interior trim towards the rear. The 24 V power converter and the serial cable, which are all provided in one wiring harness, will then be in proximity of the vehicle’s 12 V power supply and the high-performance computer. Since the sensor is located externally on the vehicle it has a weatherproofing rating of IP69K, which is included with the purchase of the Aptiv ESR. This was the second of the two sensors purchased from the AutonomouStuff dealer. The specifications of the Aptiv ESR 2.5 can be found on AutonomouStuff’s website for further investigation [56].

### 2.6. LiDAR: Ouster OS1 64 Gen1

When choosing a LiDAR, we chose to prioritize high accuracy for ground truth and a system using a minimum number of data points to lower memory storage size requirements. The Ouster OS1 mid-range LiDAR offers a good compromise between resolution fidelity, which is accurate enough to function as ground truth measurement for creating High Definition (HD) maps [12,46] as well as being cost effective. Comparing the transmitted and reflected laser results in the calculation of 4 parameters, [X${}_{i}$, Y${}_{i}$, Z${}_{i}$, I${}_{i}$] where *i* denotes the ith layer, X${}_{i}$, Y${}_{i}$, Z${}_{i}$ represents the Cartesian coordinate system in free space relative to the LiDAR [longitude, latitude, elevation], and I${}_{i}$ is the intensity of the object’s reflection. Note that having 655,360 points or more per second can make a 3D point cloud, which effectively serves as ground truth for validation of other sensors on the vehicle [22,37,57]. The resulting 3D point cloud output can be viewed by applying equations provided in the Ouster software user guide [58]. The ROS driver provided on their website [59] can also be used to see the 3D point cloud using the ROS visualization (Rviz) tool. A 3D point cloud output example, 20, is shown in Section 3.1. Their download website has many other links to help visualize the data using: Ouster’s SDK Package (PyPI), WebSLAM, and a graphic user interface (GUI) for Mac, Windows, and Linux OS [59].

The power requirement for the LiDAR is 120 V AC, which is provided from the power inverter from 12 V DC to 120 V AC. 80/20 extruded aluminum building structure was used for the mounting frame to be attached to the aftermarket roof rack of the vehicle. The LiDAR is physically mounted at the center of the vehicle on a post to ensure an accurate field of view [5,22]. An aftermarket roof rack and cross-beam were required to achieve this configuration which is consistent with other configurations in the literature [12,22,37]. The data cable runs along the roof rail and then back towards the rear cargo area. The final mounted LiDAR can be seen in Figure 8. Since the sensor is located externally on the vehicle it has a weatherproofing rating of IP68, IP69K which is provided publicly on their company website. The specifications for the Ouster OS1 LiDAR can be found on Ouster’s website for further investigation [60].

### 2.7. GNSS/IMU: Swift Navigation: Duro

When choosing a radar, we chose to prioritize durability and precision since it will be mounted externally and used for localization. For the GNSS/IMU sensors, two Swift Navigation Duro sensors were chosen, seen in Figure 9. Two of these sensors were chosen (1) to provide better overall accuracy using Real-Time Kinematic (RTK) corrections since GPS location is important, and (2) to calculate the vehicle’s differential position and thus the orientation of the Ego vehicle [61,62]. RTK is a carrier-based ranging technology that offers ranges (and subsequently locations) that are orders of magnitude more exact than those provided by code-based positioning. RTK methods are challenging. The fundamental idea is to lessen and eliminate mistakes that are typical of a base station and rover combination [63]. The Duro is a military-grade multi-band, multi-constellation centimeter-accurate GNSS when using RTK that can be done with Skylark cloud correction service or base station/rover to obtain better than 10 cm accuracy. This allows the user to keep track of the vehicle’s global position and be able to localize the AV for lane-keeping [61,64]. Swift has a ROS driver which makes it easy for software installation [65]. Swift navigation also has a console for a Windows or Mac OS installer. For quick and consistent interfacing with other sensors, we chose to use ROS and Linux OS. An output exampleof the Ego vehicle’s position around the WMU Engineering campus can be seen in Section 3.1. 

The engineering mounting requirement for the two GNSS sensors is to have them both on the centerline of the vehicle [12,22]. The power requirement for the ethernet switch to transmit the data from the GNSS module is 120 V AC, which will be provided from the power inverter from 12 V DC to 120 V AC. 80/20 extruded aluminum building structure was used for the mounting frame to be attached to the roof rack of the vehicle. The mounting brackets that attach the sensors to the sensor rack were made from aluminum plates. The wiring is the same for the rest of the roof-mounted sensor rack where the cables run back to the hatch and into the rear cargo area of the vehicle. The power was connected to the 12 V power distribution and the data were transmitted through ethernet cables which are connected through a switch to the high-performance computer. Since the sensor is located externally on the vehicle it has a weatherproofing rating of IP67, which is provided publicly on their product sheet for the ruggedized Duro. The specifications for the Duro GNSS/IMU can be found on Swift Navigation’s website for further investigation [65].

### 2.8. High-Fidelity In-Vehicle Computer: Nuvo-7160GC AI Edge Computing Platform

When choosing an in-vehicle computer, we chose to prioritize high performance while maintaining minimum power consumption when testing AV algorithms. The Nuvo computer with Linux OS was selected due to its connectivity capabilities, low cost, and low power consumption when compared to the Nvidia PX2. The Nvidia PX2 has an ARM architecture which also limits the sensor driver capability. The Nuvo 7160GC is equipped with the Nvidia Geforce 1660 Ti which can handle the processing of autonomous algorithms such as real-time object detection [66] deep learning such as convolutional neural networks [67], sensor fusion [6,10], path planning [17,68], and waypoint following [69].

The Nuvo PC also has a DC and AC power input, which is ideal for integration with the vehicle’s 12 V power source, and it supports bench testing prior to installation. The computer can have its power supply along with its very own fuse when integrated into the vehicle without the need for a high-power AC source like the Nvidia PX2. The computer, which can be seen in Figure 10, consumes 185 W of power, which is far less than most high-performance PCs with GPUs used for AV integrations. In comparison, the Nvidia PX2 consumes 250 W, thus we have achieved 65 W energy savings. The energy savings was due to the increased computational power of the PX2’s GPU, which is not a comparison of the computer power yet to show the best High Performance PC for quick integration and the OS architecture to collect data and to test algorithms for autonomous/ADAS applications. Additional discussion of system power consumption is presented in Section 3. The computer was mounted in the rear cargo area for accessibility and convenience due to the proximity of the power source as well as all the sensor connections. The computer must be turned off before the vehicle to ensure no degradation in the vehicle battery, which may result in the vehicle becoming inoperable. The specifications for the Nuvo-7160GC Edge computing AI platform can be found on Neousys-tech’s website for further investigation [70].

### 2.9. System Architecture and Hardware Integration

Once the sensors are installed in the locations shown in Figure 2 and working, the full system architecture should be evaluated for loose connections, bracket stiffness to ensure false detections from vibrations created by air drag are not present, consistent sensing results, and more. Purpose-built wiring and hardware are then put into place to ensure a professional aesthetic, as well as compliance with all power requirements. The overall system architecture diagram is shown in Figure 11 as well as the system wiring diagram, shown in Figure 12, to illustrate the proper connection for each sensor to the high performance computer.

A Dewalt 1000 W inverter was added since there was a need for conversion of DC power to AC power. The AC 120 V power is for the LiDAR and the ethernet switch. There is also a connection on the back of the inverter that can be used as a 12 V power supply. This 12 V source is used for the radar, MobilEye, Drive-Kit, and the solenoid switch to power the circuit. The radar needed a separate ground that is not a chassis ground due to signal interference, so it has a separate circuit that runs parallel to the power distribution. The radar’s power source is then converted to 24 V. The Swift Duro GNSS/IMU and the Linux Nuvo PC both run off of a 12 V power source from the power distribution fuse box.

Multiple ROSBag collections of multiple drive-cycles in varied weather situations have worked with the current software integration of the sensors. The hardware integration of the external sensors has been put to the test in a variety of weather situations, including hot, humid, rainy, and snowy. Given our lab’s location in Southwest Michigan, we are subjected to a wide range of weather conditions. The Ego vehicle has also been put to the test in touchless car washes with high pressure systems while the vehicles hardware still continues to work as expected. The vehicle was also tested at the American Center for Mobility (ACM) [71] using their raining simulation truck while testing our MobilEye/Aptiv ESR dynamic region of interest sensor fusion, which helps reduce raw radar detection by fusing lane features with raw radar detection, reducing raw radar detection by up to 100% [72]. The lab has demonstrated how extensive the system integration is and how it can perform in poor weather circumstances after 2000 miles and over 1000 h of experiments and hardware operations in varied weather situations.

## 3. Results

Once the vehicle, components, and sensors have been chosen and installed, system and sensor results can be obtained. In this section, we will demonstrate the successful operation of all sensors, total system cost, total system energy use, as well as general observations.

### 3.1. Sensor Performance Validation

To validate sensor performance, the sensor outputs are compared to LiDAR detections and human evaluation, which both serve as the ground truth. Sensor ground truth validation was accomplished for radar object detection, MobilEye object detection, and MobilEye lane line detection. We were able to observe lane lines with the LiDAR and identified several instances of ghosting that were eliminated by counting the misdetections and distances of those objects [5,35,36]. Since the Aptive ESR only detects 64 objects at a time, the ZED, MobilEye, and Ouster LiDAR will be used as another form of object detection. In general, this process is required for research in sensor validation, sensor fusion, perception models [73,74], and controls, in the applications of improved safety and energy efficiency [75,76,77]. The system over stability has been reliable during multiple test during inclement weather conditions such as snow, rain, and heat. The system network of sensors all work as specified by the manufacturer, and the ROS driver found all have consistent rates with low latency. This newly instrumented platform does not have V2X 4G/5G capabilities, yet has the ability to be added to the system. Note, that when adding a modem for LTE, services take in consideration the latency of the network provider, as well as multiple service providers to prevent loss in services [78]. Our lab plans on accomplishing this as future work for teleoperations in collaboration with DriveU [79].

Our lab has already dove deep into research using our newly instrumented Kia for better understanding of waypoint following using pure pursuit [80] and then checking the resilience of our discovery of waypoint controllers [81]. We have also done research with LiDAR and cameras in various weather conditions to have a better understanding of how our new platform will perform [82]. Weather circumstances were stated in Section 2.9, highlighting how sensor fusion between the MobilEye and radar can help with high accuracy and low computational power [72], which can be seen in Figure 13. This method was also demonstrated at The American Center for Mobility (ACM) for Destination ACM 2021 in Michigan [71]. This demonstration shows that it works well in heavy rain conditions, and can be seen in Figure 14, once again showing its versatility. Literature of how to use snow tracks in snowy conditions to compute lane attributes for lane assist algorithms [83] has also been done by the lab, which further shows the possibilities for the low cost AV research platform.

Raw GNSS data has typically an accuracy of 5 m; therefore, it requires RTK corrections to correct for common errors on satellite navigation systems. Therefore, the Skylark correction cloud service from SwiftNav can be purchased so the GPS sensor receives the appropriate GNSS corrections. This cloud service provides the necessary RTK corrections to achieve an accuracy <±15 cm. To reduce this accuracy even further, the loosely coupled fusion model the Duro sensor possesses was activated. This fusion model combines the GNSS data and data from the integrated IMU sensor to obtain an accuracy <±10 cm.

A confidence ellipse [84] was used to evaluate the accuracy and the variance of the data in both axes. A static test was performed to evaluate the positioning accuracy of the GNSS sensor without corrections (raw), with RTK corrections and with the RTK + Loosely coupled fusion. The confidence ellipse tells us the region for predicting where new data falls into. Results of our static test can be seen in Figure 15. One standard deviation means 68% confidence the data will fall within this region, two standard deviations means 95% confidence the data will fall within this region and three standard deviations means 99% confidence the data will fall within this region. As you can see in Figure 15, the raw GNSS data contains a lot of variance in both variables; nevertheless, as we apply the RTK corrections and increase the complexity of the model, the spread of the data points reduces significantly. Therefore, there is more variance in the *y*-axis than the *x*-axis (this is typically the case). Additionally, when using the RTK corrections + Loosely coupled we can see the data has a standard deviation of approximately 5 cm in the *x*-axis and 4 cm in the *y*-axis compared to 5 cm in the *x*-axis and 6 cm in the *y*-axis with just the RTK corrections.

Even though the SwiftNav sensors report the horizontal and vertical accuracy when in RTK mode or RTK + Loosely coupled mode, the WMU and its team verified the incoming GNSS data using some statistical analysis by performing some stationary tests using GNSS data collected from the Trimble as ground truth. The Trimble is a GNSS receiver capable of achieving an accuracy <±1 cm. This was performed by leaving WMU’s research vehicle stationary at the Engineering Campus parking lot, mounting the Trimble [85] on top of the vehicle, collecting the GNSS data, swapping back the SwiftNav sensor, and inputting this data as the local origin so the sensor calculates its position with respect to that point.

### 3.2. Sensor Operations

The Drive-Kit provided from Polysync was integrated in parallel to the current OEM system without splicing, which made integration effortless. Since the Drive-Kit has its own CAN bus system, it can communicate (in parallel) with the vehicle’s CAN bus system so as to not flood the OEM CAN bus, which can cause faults in the vehicle controls. There is some need to test the steering and throttle command to tune them correctly before using the Drive-Kit with complex AV applications. The Drive Kit’s measured minimum power consumption is 2.7 Watts and has a cost of $60,000.

Figure 16 shows the RGBA image output as well as the depth image output for the ZED stereo camera. These outputs can be used for the development of custom computer vision algorithms and/or custom sensor fusion algorithms when developed with open-source data sets such as the KITTI dataset [46,86]. The camera has a total power consumption of 1.9 Watts and a cost of $449 [47].

Figure 17 shows the MobilEye 630 detections from the forward-facing detection of lane lines, edges of the road, and vehicles. The lane line detections are the dashed and solid green lines while the edge of the road is the red line. The green lane lines indicate high confidence in the lane line detection, whereas the edge of the road will always be the color red as a warning color. As the confidence decreases, the color of the lanes will change in color from green (high), yellow (mild), and then red (low). The blue boxes show the detected object and its distance as well as its classification. These outputs are currently in use by OEM’s in their current ADAS [49]. The LiDAR and RGB camera was used to validate the objects detected from the MobilEye to ensure that the objects are in the correct vector space. The total power consumption of the MobilEye 630 is 5.2 Watts with a cost of $11,900 [87]. 

The Aptiv ESR detections can be seen in Figure 18 as green boxes. These green markers are the detection’s provided through the ROS driver for any object in the field of view, but unlike the MobilEye, it does not classify objects. The tracked detection has a large reduction in detection compared to the raw detections. Figure 19 illustrates the raw detection in a short amount of time, as well as the profile of the radar fan FOV for both the long and short range radar packages. If learning techniques are used to help classify detection using the Radar Cross Section (RCS), be sure to use the raw CAN data to ensure no loss in information through the ROS driver. There is still a means for data processing that is required to stop cluttering, ghost detections, and object track selection [51,88]. The LiDAR and RGB camera were used to validate the objects detected from the ESR to ensure that the objects are in the correct vector space. The total power consumption of the Aptiv ESR is 18 Watts with a cost of $2500 [88]. 

In Figure 20, a 3D point cloud can be seen for the output of the Ouster OS1 LiDAR using Rviz functionality in ROS. The output shows the vehicle driving on the highway where the lane lines can be seen as well as the barriers on both sides of the road. The point cloud shows the intensity of the reflections with a blueish color. The red colors are the less intensive reflection. Notice that the further away the point, the less reflective its value. The LiDAR’s reflection intensity can help with object and lane line detection [41,89], and also help to develop high-definition maps for AV research [90]. Figure 21 demonstrates occlusion in the LiDAR point cloud, which includes a car to the left of the Ego vehicle, a bus to the front left, and a fence to the right, all of which have occlusion behind them. The total power consumption of the Ouster’s OS1 is 20 Watts with a cost of $8000 [91].

Figure 22 shows the Swift Navigation GNSS/IMU sensor outputs of vehicle location. With this information, researchers can obtain an accurate heading and localization of the vehicle. This can lead to rapid and sound technology demonstrations built upon waypoint following. Using known controllers such as the Stanley or Pure Pursuit (in highway or city driving) can rapidly increase autonomous capabilities which are currently being used by researchers [92,93]. The individual power consumption of the GNSS/IMU and GNSS are both 5 Watts with a cost of $4995 [94], and the GNSS without the IMU costs $3495 [95].

In order to have an accurate vector space of the Ego vehicle’s environment, all sensors require transformations from one sensor to the next [96]. This is done in ROS and allows the sensors to accurately display the perceptions in Rviz. The only sensor that needs to be installed and calibrated is the MobilEye which is recommended by the manufacturer and AutonomouStuff. The rest of the sensors work with the correct accuracy out of the box and with the drivers shared on GitHub which can be found in Appendix A. All external sensors are waterproof IP6X rated with military and industrial grade connectors, yet there can be special consideration to add an extra protective layer to prevent corrosion at the connectors. Additionally, take in consideration how the sensor is enclosed to prevent occlusions.

### 3.3. System Cost

The 2019 Kia Niro base vehicle cost is $23,450 MSRP [97] and the Drive-Kit costs $60,000, which was the majority of the cost for AV instrumentation. Note that the Drive-Kit was the most expensive component of the entire vehicle integration. We felt that this cost was justified since it saved significant amounts of time in CAN signal decoding with the use of a Database Container (DBC) file and CAN operation using the ROS drivers provided by Polysync. The time saved increased our research capabilities for autonomous control of the vehicle platform. Future products in this space are anticipated to provide the same level of functionality for a much reduced cost [28]. The DBW vehicle and Drive-Kit made it ready to be retrofitted with sensors for AV applications. The completed cost-effective AV research platform can be seen in Figure 23, and Table 2 shows the budget and power consumption for each system component as well as the total. The total cost of building this AV research platform is $118,189 in 2020 USD. Note that the DBW subsystem was the most expensive, but in our opinion, this is a safety critical system that must be sound. Note that currently available AV research platforms can cost more than double.

### 3.4. System Energy Use

The total energy consumption is 242.8 Watts, which is comparable to other studies in the literature [19], showing that there is consistency when integrating an array of sensors to a DBW platform for AV applications. Keeping the power consumption at a minimum is crucial when using HEV and/or EV platforms. The greater the power consumption, the lower the range (miles per gallon equivalent, MPGe) of the vehicle. The need to maximize the efficiency of the vehicle has become more relevant since EVs are still fairly new to the market and research is still underway to optimize the range with normal operation without autonomous control [98]. Current research for increasing fuel economy for HEV/EVs with autonomous features have been considered such as increased ADAS [76], as well as sensors from autonomous platforms that communicate with the infrastructure as well as onboard detection systems for lead vehicles [99]. Our lab has current research for energy efficiency autonomous vehicle, which involves Neural Networks for velocity predictions and high-fidelity modeling [100,101], Machine Learning for velocity predictive optimal energy management [93], and optimization for autonomous eco-driving with traffic light and lead vehicle constraint for optimal trajectory of energy management in Battery Electric Vehicles (BEV) [99]. The need has grown to optimize the sensor suite packages in order to have a minimal effect on the range with HEV/EVs for autonomous applications.

### 3.5. Results Summary and Discussion

Finally, while there are several approaches that can save integration time, such as [19,20,21,22,23], none of them have demonstrated energy use and cost of the AV system in their studies. First, sensors under consideration should be benchmarked to quantify their operational and start-up power consumption as this will affect which fuses to use. Second, the use of ROS greatly speeds up system integration as has been shown in other research [16,17,18,80]. Third, verification that all sensor drivers and the Linux OS are installed and operational before testing saves significant debugging time. The correct sensor drivers can be installed using a bash file given on GitHub by WMU’s Energy Efficient Autonomous Vehicle (EEAV) Lab for convenience of use by the researcher. Next, cable routing and power use for each sensor should be planned (see Section 2). Lastly, make sure to use a switch with a high bandwidth rating for the ethernet connections. The ZED camera does benefit from a housing that surrounds the camera and is as close to the windshield as possible to avoid any glare. Our lab has made a 3D prototype which shows great results in reducing glare and distortion in the images. The radar must also be connected to its own ground and not a chassis ground to avoid unwanted noise. The inverter will also stop working if the vehicle battery is low and will affect the LAN switch as well as the Ouster OS1 LiDAR.

## 4. Conclusions

Herein, we demonstrate the development of a cost-effective research platform. This fills a research gap of a comprehensive study detailing sensor selection, performance, power consumption, and cost considerations, and will act as a reference for future studies of energy efficient AVs. Detailed discussion for sensor choice was presented on why they were selected (performance, power consumption, and cost), how they were instrumented, and their outputs for each sensor and their corresponding system components. Our results show successful sensor outputs from all integrated sensors (cameras, LiDAR, radar, GNSS, and IMU) along with the computer control ability of the vehicle using DBW. Our proposed novel research shows how to instrument a DBW platform by showing the performance, power consumption, and cost of the sensors while also including ROS driver for a more straight-forward instrumentation for converting a DBW into an AV research platform. The total power consumption of this vehicle is 242.8 Watts and the overall cost is $118,189 USD. The power usage is comparable to current research, and the equipment is significantly less expensive than purchasing a complete sensor and software package.

Overall, this research addresses sensor selection, performance, and, more importantly, power consumption, as well as cost considerations, demonstrating that the development of an AV research platform is becoming more straightforward and cost effective, which is critical as researchers previously devoted to engine control are now exploring control of autonomous electric vehicles as the automotive engineering discipline evolves. It is our hope that this study, shown in Table 2, includes crucial information that is needed to further the development in this field, and to inspire researchers who have yet to conduct real world testing to make the transition. The limitations of this platform is (1) the redundant 360 degree view with camera and only with the use of a LiDAR, (2) currently does not have a LTE modem for V2X communications. Even with the limitations of the current platform, it is still an intermediate step to get researchers more involved in the field. The platform was designed to be able to add/swap different sensors and equipment. In future work, our team plans to use this AV research platform to develop sound sensor fusion, sound AV operation in inclement weather, energy efficient AV operation, and AV operational resilience. The lab currently received a new Department of Energy infrastructure grant to build another AV platform using a fully electric vehicle to study how AV sensor suites and algorithms affect battery capabilities while using the infrastructure sensors and LTE services to help reduce the computational power needed for AV. Using this paper as a reference, this paper will serve as a starting point for the ease of manufacturing AV platforms. Additionally, follow-on studies are needed from the research community that compare various AV research platform builds using our proposed energy consumption and cost analysis; this will become increasingly important since AV sensor technology continues to evolve at a rapid rate. Overall, as evidence of effective strategies for realizing safe and sustainable transportation emerges, our research community can progress this research field closer to widespread commercialization.

## Figures and Tables

**Figure 1 sensors-22-05999-f001:**
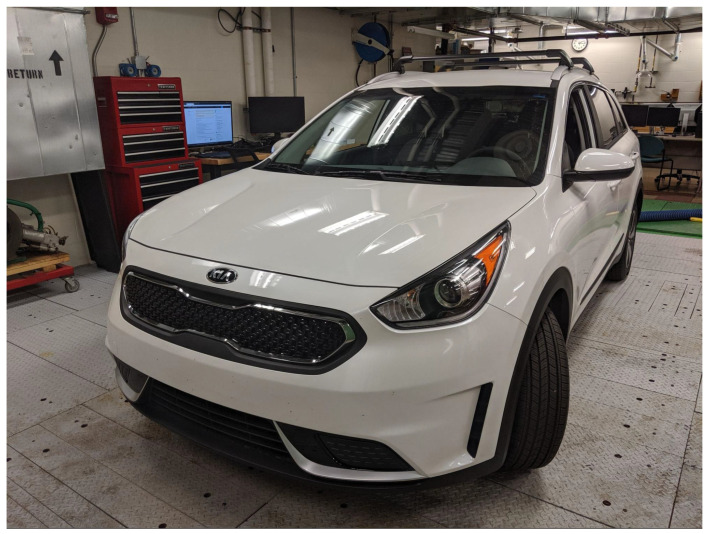
Stock 2019 Kia Niro, DBW compatible, as purchased from a local Kia dealership.

**Figure 2 sensors-22-05999-f002:**
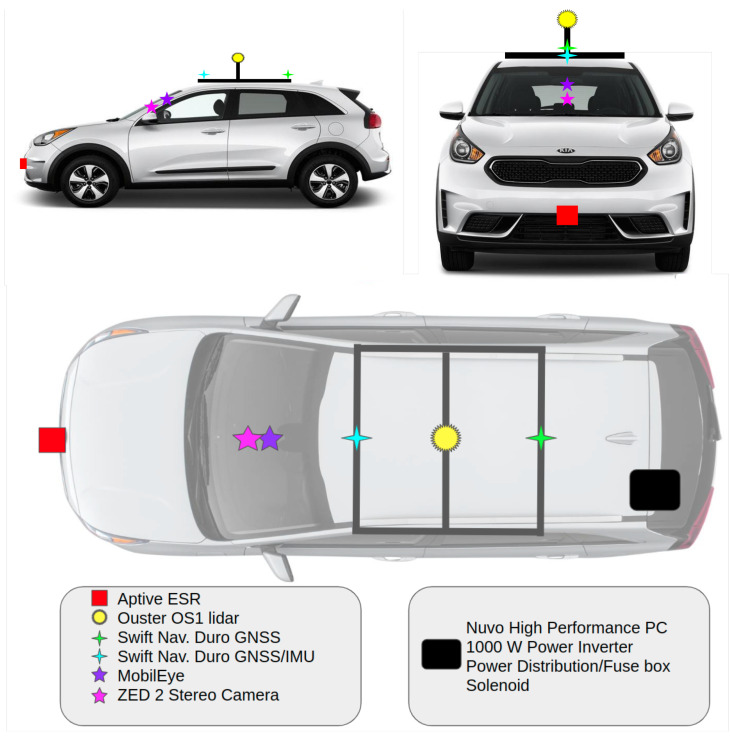
Sensor and hardware placement for autonomous integration side, front, and bird’s-eye view. Image sourced from [43].

**Figure 3 sensors-22-05999-f003:**
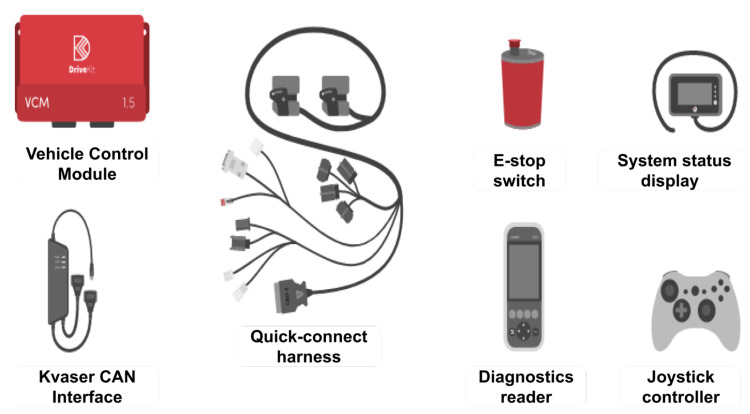
Polysync Drive-Kit with all of the components listed by name that are needed for installation.

**Figure 4 sensors-22-05999-f004:**
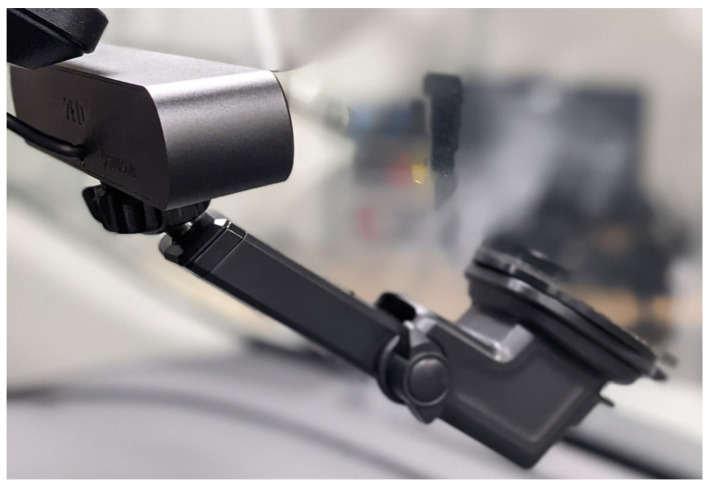
Mounted ZED stereo camera used for forward-facing computer vision.

**Figure 5 sensors-22-05999-f005:**
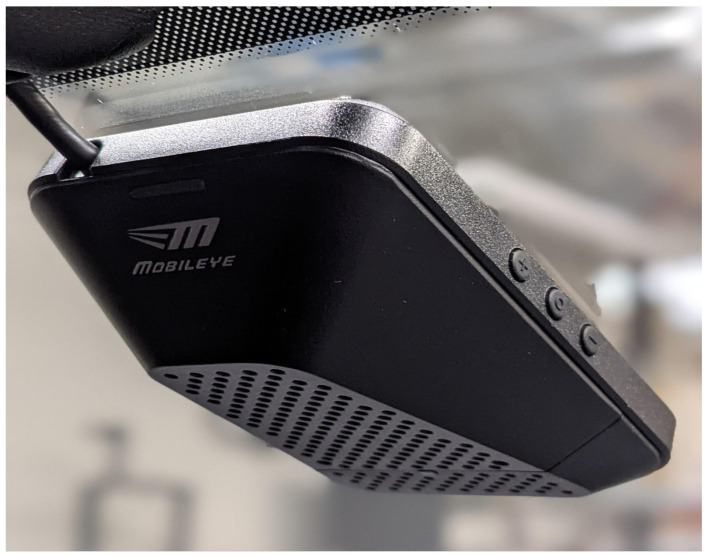
Sensor and hardware placement for autonomous integration.

**Figure 6 sensors-22-05999-f006:**
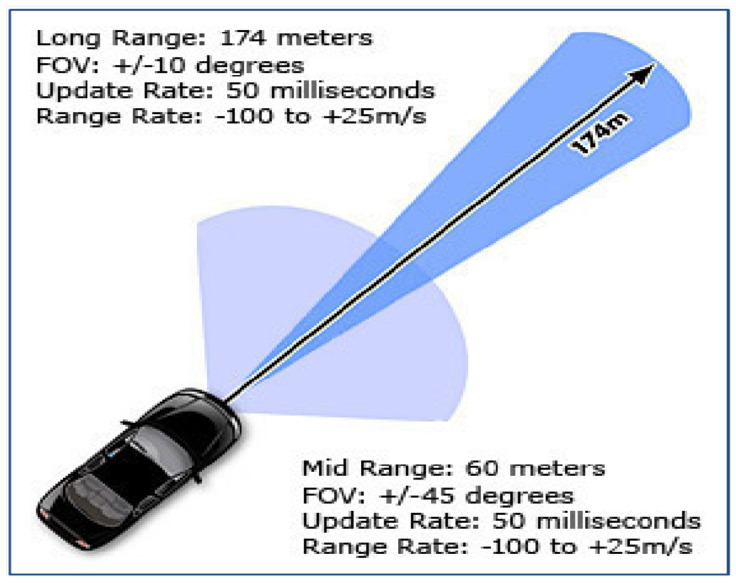
Multiple fields of vision zones for the Aptiv radar for increased safety measures [53].

**Figure 7 sensors-22-05999-f007:**
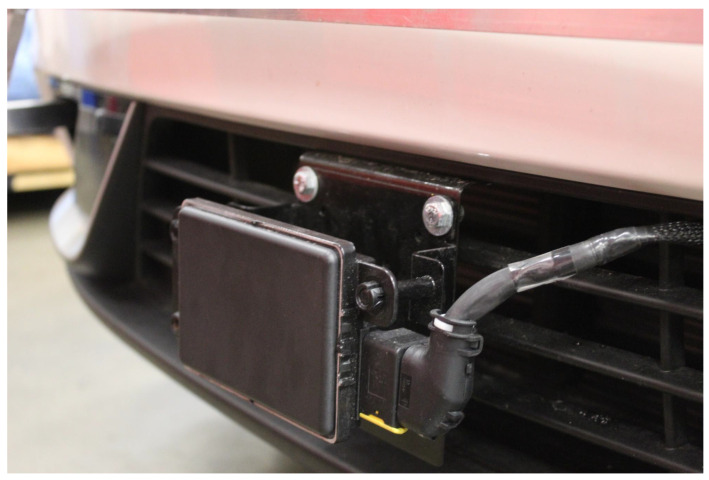
The bracket is mounted to the bumper behind the bumper cover of the vehicle for structural integrity.

**Figure 8 sensors-22-05999-f008:**
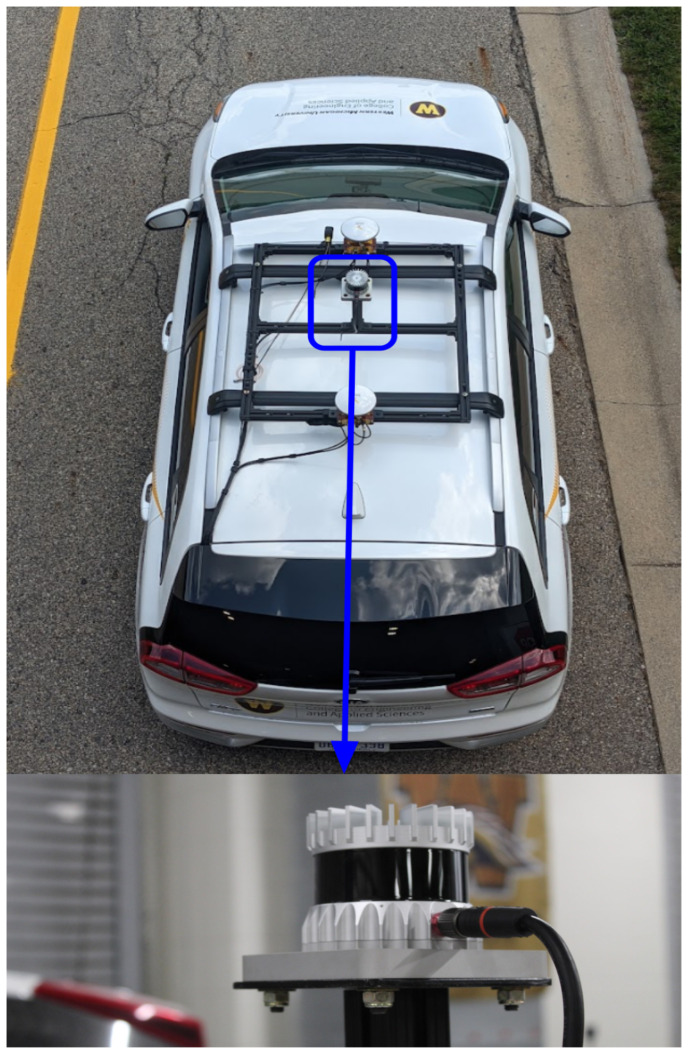
Ouster LiDAR OS1 mounted near the center of the vehicle with proper height for a surrounding view of the platform.

**Figure 9 sensors-22-05999-f009:**
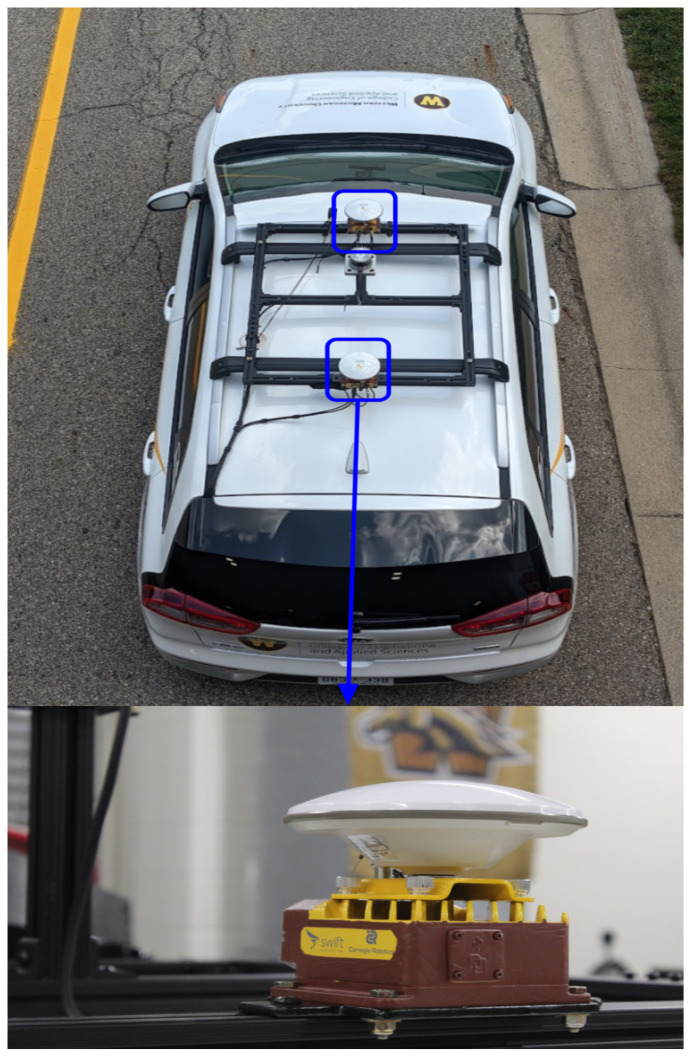
One of the Swift Navigation Duro sensors fitted onto the roof’s mounting rack (painted) with the offset of the heading configuration is set to zero degrees.

**Figure 10 sensors-22-05999-f010:**
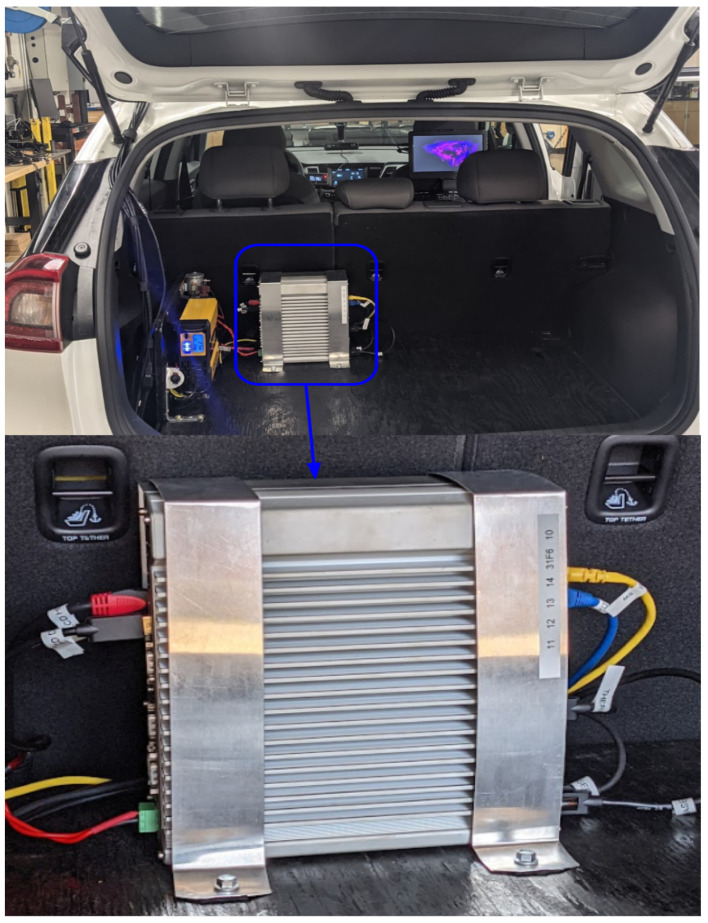
Nuvo PC—Ruggedized GPU-Computing Platform Supporting 120 W Nvidia^®^ GTX 1660Ti GPU and Intel^®^ 8th-Gen Core™ Processor, including i7-8700 pre-installed with Linux 18.04. Mounted in the rear of the platform.

**Figure 11 sensors-22-05999-f011:**
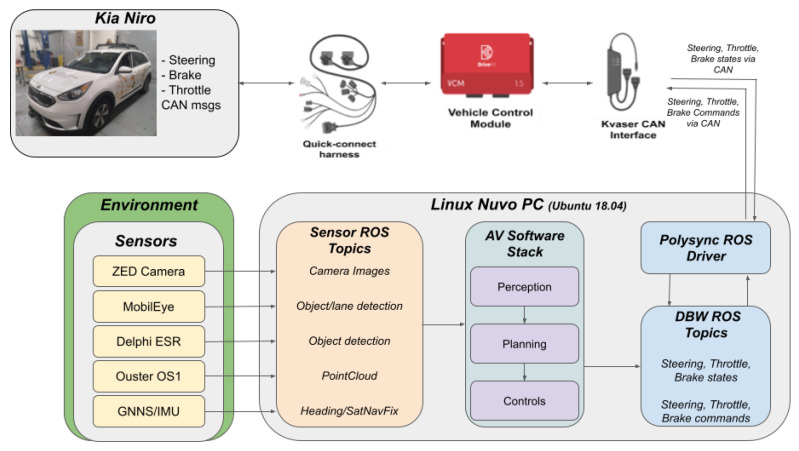
Sensor system diagram for the Ego vehicle Kia Niro platform. The flowchart shows the interaction with the vehicle, integrated sensors, Drive-Kit, the autonomous stack, and the flow of information between them all.

**Figure 12 sensors-22-05999-f012:**
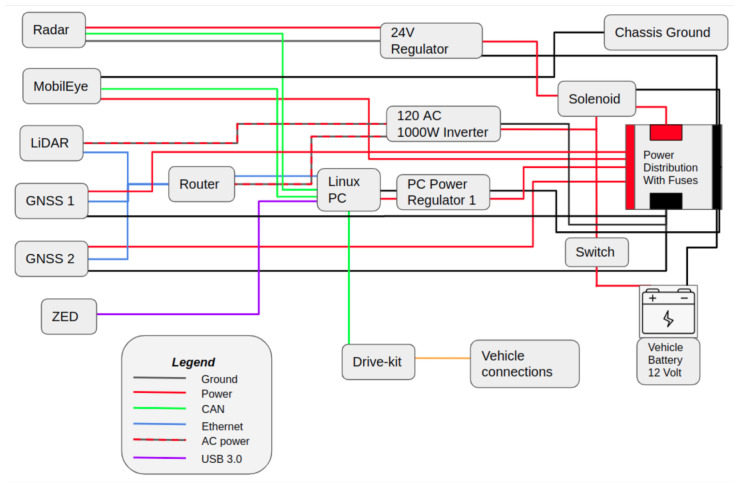
Wiring diagram showing the power connection as well as the type of connection for each sensor and hardware component.

**Figure 13 sensors-22-05999-f013:**
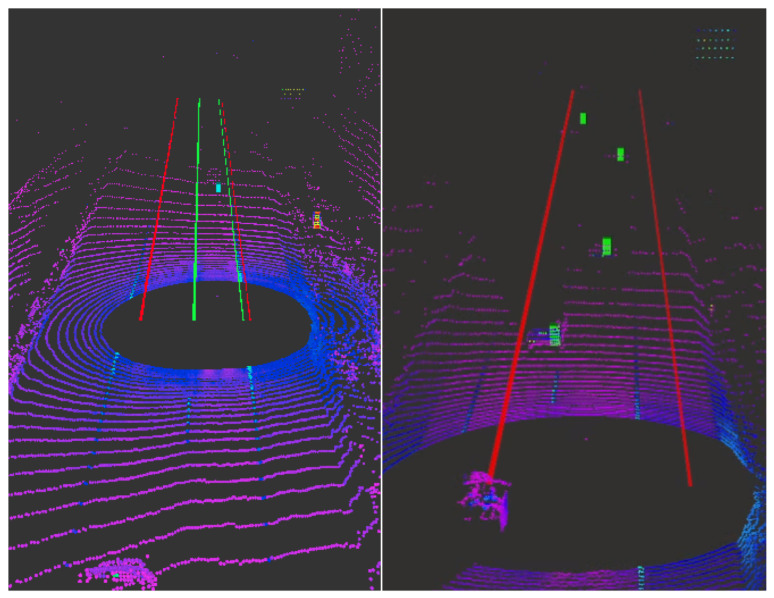
Higher accuracy and low computational power for MobilEye/Aptiv ESR sensor fusion using a Dynamic Region of Interest (DROI) [72]. Current sensor fusion algorithm to reduce radar detections, which can be seen as the blue marker for the (**left**) and green markers for the (**right**).

**Figure 14 sensors-22-05999-f014:**
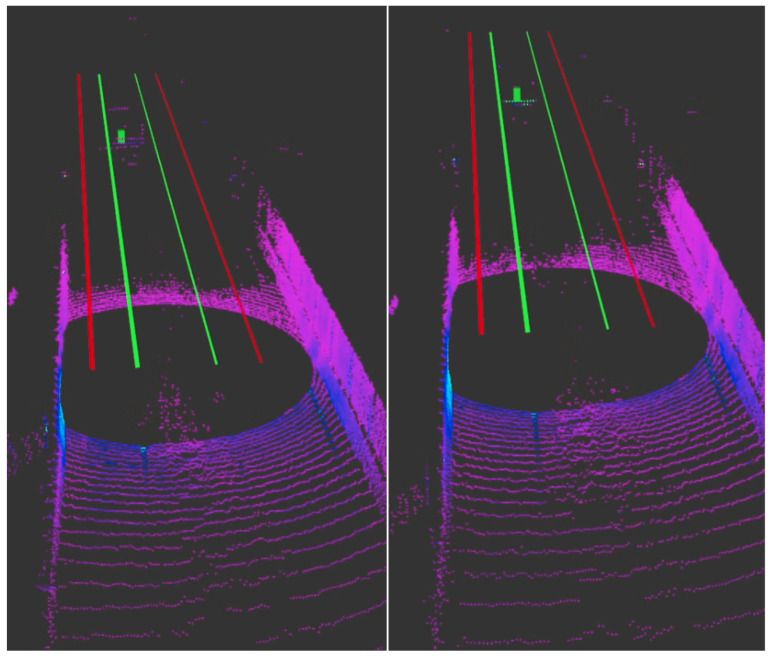
Demonstration of the sensor operations and sensor fusion [72] in heavy rain condition with rain from the environment plus the rain simulated from the ACM’s rain truck [64].

**Figure 15 sensors-22-05999-f015:**
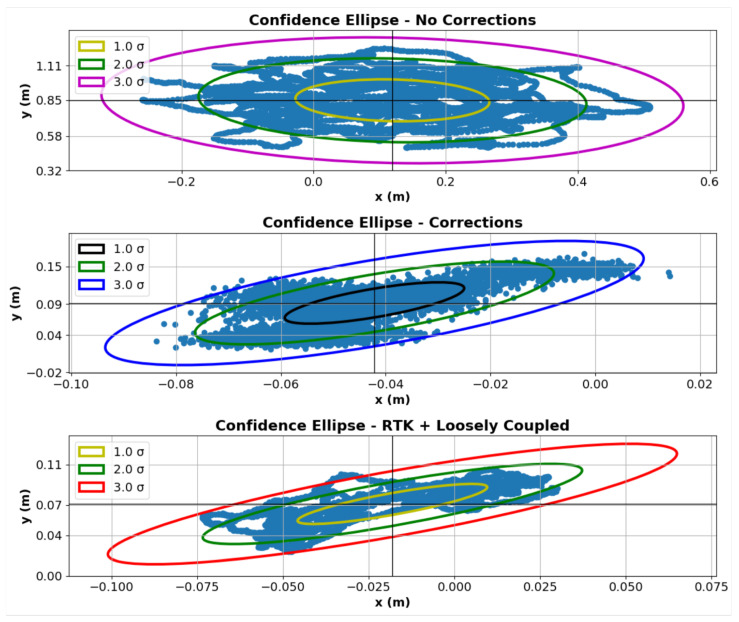
Confidence ellipse for GNSS data with no corrections (raw), RTK corrections and RTK + Loosely coupled fusion.

**Figure 16 sensors-22-05999-f016:**
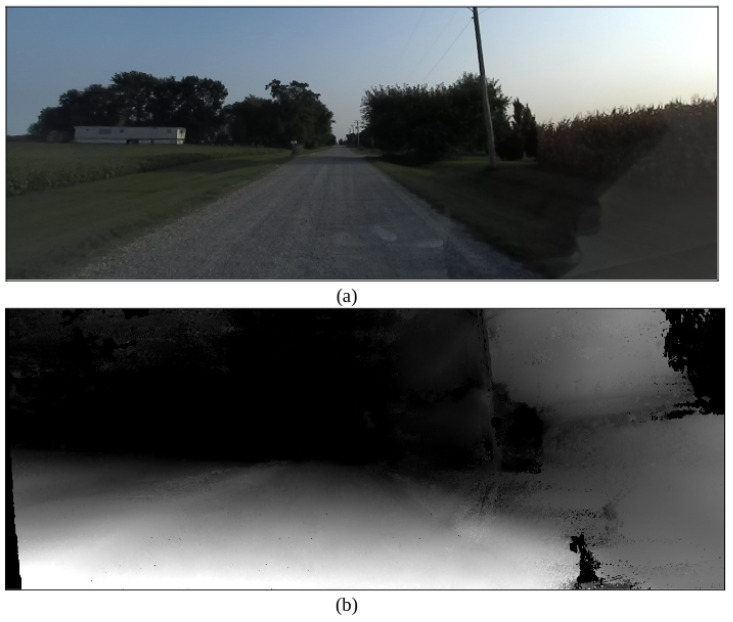
ZED camera RGBA image output (**a**) and depth output (**b**) during low light conditions.

**Figure 17 sensors-22-05999-f017:**
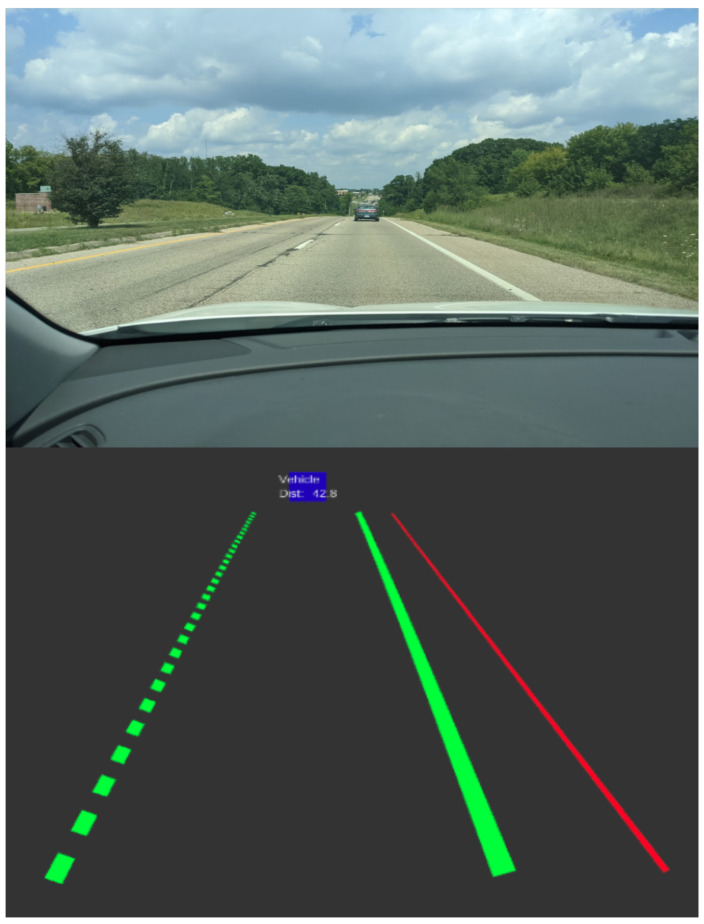
MobilEye lane line, road edge, and object detection visualization using Rviz with ROS.

**Figure 18 sensors-22-05999-f018:**
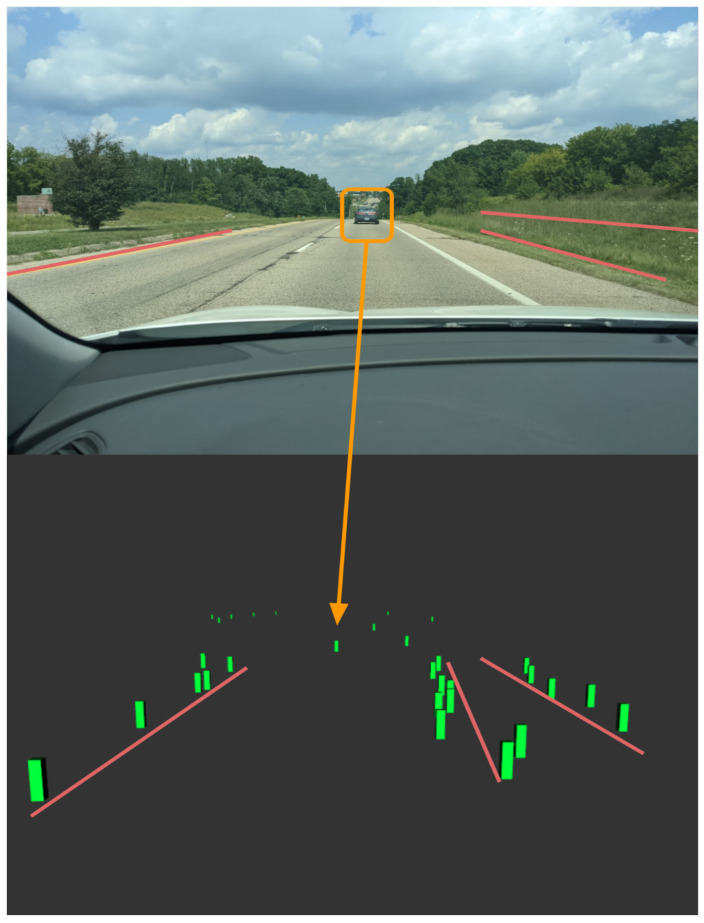
Aptiv ESR object marker visualization using Rviz with ROS. The red lines show the curb to the left and the high grass just before the berm, which is all detected by the radar.

**Figure 19 sensors-22-05999-f019:**
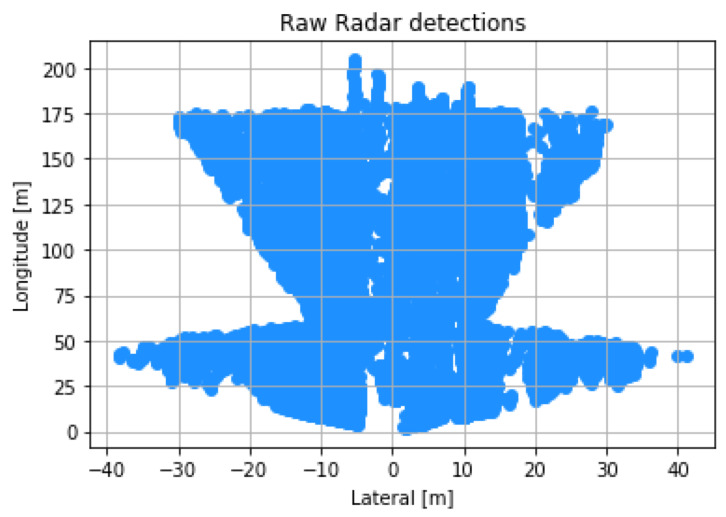
Radar distance validation. The plot shows detection over a driver cycle around WMU’s engineering campus, which includes highway driving.

**Figure 20 sensors-22-05999-f020:**
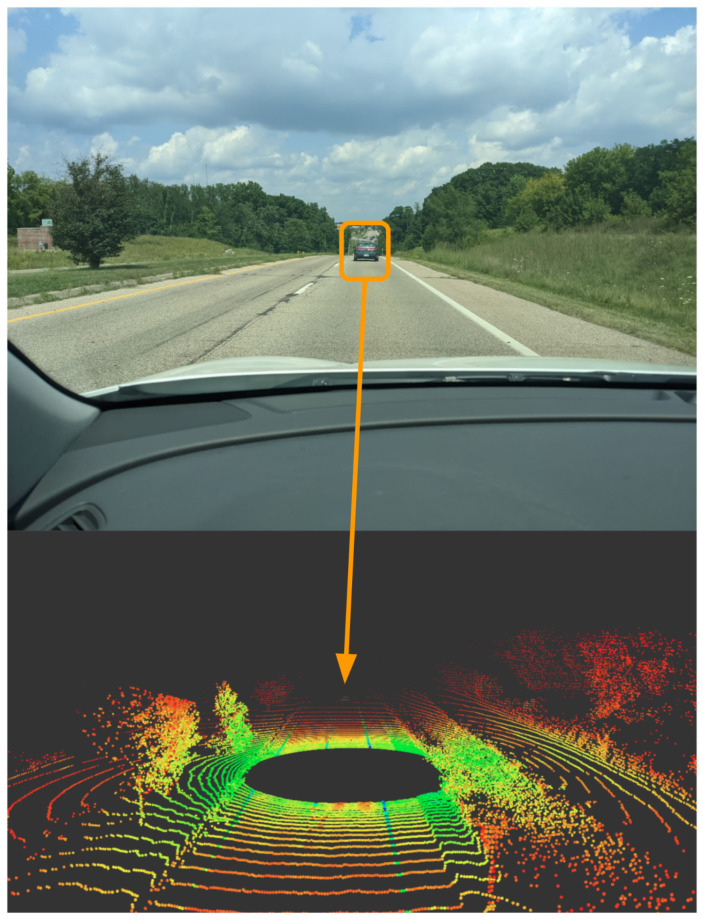
Ouster LiDAR visualization using Rviz with ROS.

**Figure 21 sensors-22-05999-f021:**
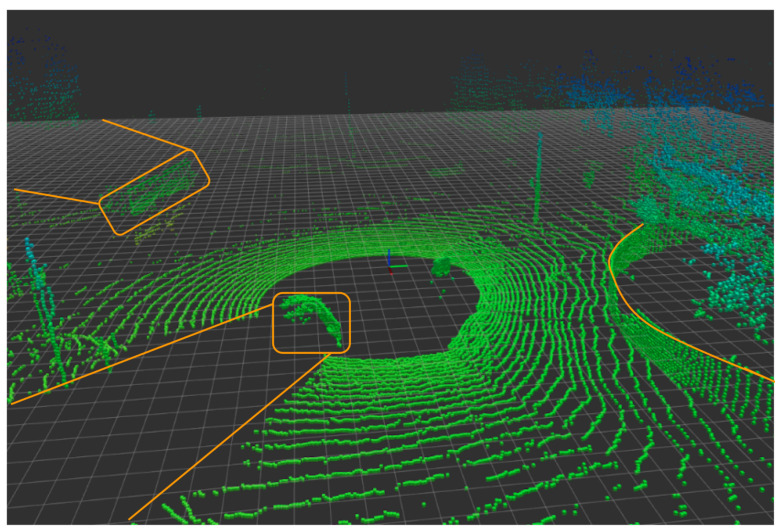
LiDAR point cloud occlusion from nearby vehicles. Three occlusions can be seen outlined with the orange lines, car, bus, and fence.

**Figure 22 sensors-22-05999-f022:**
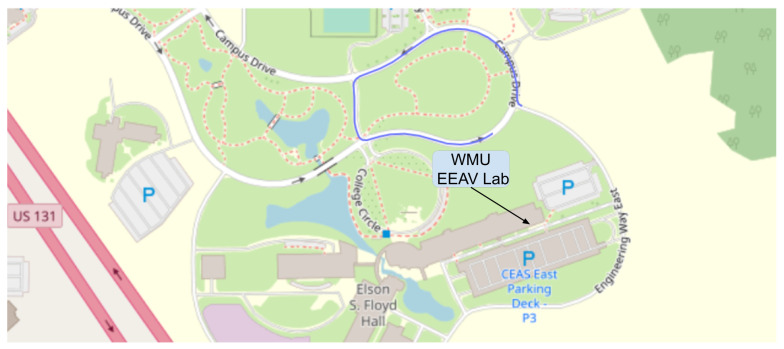
Live data of the Ego vehicle’s position (latitude and longitude) using the Swift Navigation Duro, displayed using OpenStreetMap.

**Figure 23 sensors-22-05999-f023:**
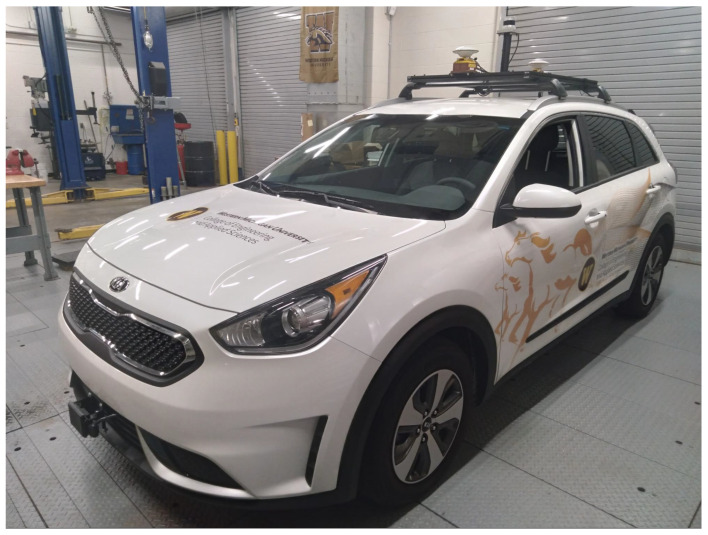
Instrumentation complete for 2019 Kia Niro.

**Table 1 sensors-22-05999-t001:** Comparison of the open source documentation, the compatibility, the cost, and the power consumption of researched sensors. The symbols represent the information gathered to max out at three symbols showing low (★), mid (★★), and high (★★★).

Instruments	Open Source	Compatibility	Cost	Power Consumed
**Camera**				
ZED Stereo Camera	★★★	★★★	★	★
Roboception RC Viscard	★★★	★★	★★	★★★
MobilEye 630	★★★	★★★	★★★	★
Carnegie Robotics S7	★★★	★★★	★★	★★★
**Radar**				
Aptiv ESR 24 V	★★★	★★★	★	★★
Bosch OHW	★	★★	★★	★★
**LiDAR**				
Ouster OS1 64 gen1	★★★	★★★	★	★★
Velodyne HDL-64E	★★★	★★★	★★	★★
Velodyne Velarray	-	-	★★★	★
Luminar Hydra	-	-	★★★	★★
**GNSS/IMU**				
Swift Nav. Duro	★★★	★★★	★	★
Novatel PwrPak	★★★	★★★	★★★	★
Trimble RTX	★	★★	★★★	★
**Drive-Kit**				
Polysync	★★★	★★★	★★	★
DataSpeed	★	★★★	★★★	-
**PC**				
Nuvo-7160GC	★	★★★	★	★★
Nuvo-6108GC	★	★★★	★★	★★★
PX2	★	★★★	★★★	★★★
Crystal build platform	★	★★★	★★★	★★★

**Table 2 sensors-22-05999-t002:** The total cost of the sensors and hardware. (-) indicates that power was not considered. * The costs used were all found via internet search engines (based on end of 2019 pricing).

Instrumentation	Power Consumed	Cost
2019 Kia Niro HEV	-	$23,450
ZED Stereo Camera	1.9 W	$449
MobilEye 630	5.2 W	$11,900
Aptiv ESR 24 V	18 W	$2500
Ouster OS1	20 W	$8,000
Duro Starter Kit	5 W	$3,495
Duro Interial Starter Kit	5 W	$4995
Polysync Drive-Kit	2.7 W	$60,000
Nuvo-7160GC	185 W	$2900
Hardware/Supplies	-	$500
**Total**	**242.8 W**	**$118,189**

## Data Availability

Not applicable.

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
