# Peer review of "Development of an Energy Efficient and Cost Effective Autonomous Vehicle Research Platform"

_sensors, 2022, doi:10.3390/s22165999_

Round 1

Reviewer 1 Report

In this paper, a cost-effective R&D platform is developed, which fills the gap in the field of comprehensive consideration of sensor cost and efficiency, and provides a reference for future energy-saving Autonomous Vehicle research.

1. The introduction part is suggested to be appropriately deleted, and the space is a bit too long, so it can be simplified.

2. When introducing the layout of the sensors, it is recommended to add the front view and left view of the car so that the reader can clearly see where the sensor is located.

3. If the number of objects around the car is more than the number of objects that can be detected by the radar, is there any backup measures?

4. In order to save memory, choose a radar with a lower configuration. Will this affect the driving safety? If there is an impact, it is recommended that the author add some solutions.

5. In addition to the sensor itself needs to have a certain waterproof function, it is recommended that the author consider adding protection measures outside the sensor.

6. RTK is mentioned in the text, it is recommended that the author add an explanation of RTK.

7. Robust is mentioned in the text. It is recommended that the author add an introduction to robust to facilitate readers' understanding.

8. When using the HEV platform, is it reasonable to take the lowest power consumption? Authors are advised to check here to find an optimal value.

9. It is recommended to add measures to block light at the ZED camera, instead of letting it simply lean against the windshield.

Author Response

Thank you for taking the time to review my paper. The comments you made are important to the paper to help make it high level for future readers. 

Thank you, 

Nic

Reviewer 2 Report

This paper develops an efficient (in terms of cost and energy) methodology for researching autonomous vehicles. The methodology proposed is innovative and useful. However, there some comments must be addressed before being considered for publication.

Introduction, the literature review is too general, it only briefly described the existing AV research platforms. However, what are the key strengths and limitations of the platforms mentioned?

Also, more in-depth review of relevant previous research and studies should be added, for example, add more review about pervious research regarding development and evaluation of automation/automated vehicle testing and research platforms and discuss what are key performance indicators adopted by previous research for evaluating an automated vehicle research platforms?

Table 1 shows the ranking results of different sensors from four dimensions-open source, compatibility, cost and power consumed. However, it is not clear what are the ranking rule here? For example, how much cost is ranked one star (low)? And how much should be ranked three star (high). A clear rule must be provided.

Results

When validate the designed platform, have you considered system stability? What about the network latency?

Has the platform been tested using both 4G and 5G networks? Is there any latency issue?

Table 2 provided the power consumed and cost of the system. However, these figures has not been compared with those of the previous platforms mentioned in the introduction (literature review). Without such comparison, how would you convince the reader that the proposed platform is ‘efficient’ in terms of cost and energy?  In the comparation, statistical tests should be used.

Conclusion

The conclusion should be revised and improved from the following aspects:

firstly, the novelty of the proposed platform in terms of system design and system structure should be added.

Secondly, total energy and cost savings compared to using the existing automated vehicle research platforms should be added.

In addition, the limitation of the research should be added.

And finally, the implication of the research/proposed platform to automated vehicle manufacturing and development, as well as the adoption should be added. 

Author Response

Thank you for taking the time to review our paper to help make it a high-quality research paper that will hopefully encourage more researchers in AV development. 

Thank you, 

Nic

Round 2

Reviewer 2 Report

Thanks for revising the paper. The quality of the paper has been improved. No further comments from me.